# OUTCOME-DRIVEN ACTION FLEXIBILITY FOR ROBUST OFFLINE REINFORCEMENT LEARNING

## ABSTRACT

We address the challenge of offline reinforcement learning using realistic data, specifically non-expert data collected through sub-optimal behavior policies. A primary concern is that the learned policy must be conservative enough to manage *distribution shift* while maintaining sufficient flexibility for generalization. To tackle this issue, we introduce a novel method called Outcome-Driven Action Flexibility (ODAF), which seeks to reduce reliance on the empirical action distribution of the behavior policy. Specifically, we develop a new reward mechanism that evaluates whether the subsequent states, following the current policy, meet specified performance requirements (e.g., safety—remaining within the state support area), rather than solely depending on the characteristics of the actions taken (e.g., whether the action imitates the behavior policy). Besides theoretical justification, we provide empirical evidence on widely used D4RL benchmarks, demonstrating that our ODAF method, implemented using uncertainty quantification techniques, effectively tolerates unseen transitions for improved "trajectory stitching," while enhancing the agent's ability to learn from realistic non-expert data.

## 1 INTRODUCTION

Offline reinforcement learning (RL) aims to learn a high-capacity policy from an offline dataset previously collected via a behavior policy (Zhang & Tan, 2024), which has yielded significant improvements in various fields, including robotics tasks (Mnih et al., 2015; Peng et al., 2017), game playing (Silver et al., 2017), and large language models (Achiam et al., 2023; Touvron et al., 2023). However, prior studies (Fujimoto et al., 2019; Kumar et al., 2020) have indicated that offline RL algorithms suffered from the *distributional shift* (Fujimoto et al., 2019; Jin et al., 2021b) problem, where the divergence between the new and behavior policies makes the agent encounter with some unseen actions or states, which are challenging for generalization.

In practice, obtaining ideal expert data is often challenging, and the most realistic data used for training is generated through sub-optimal behavior policies. The difficulty of addressing *distributional shift* becomes more pronounced when learning from realistic non-expert data, as blindly cloning these potentially highly sub-optimal behaviors can be dangerous. Unfortunately, many previous works, such as Behavior Regularized Actor-Critic (BRAC) (Wu et al., 2019b), Conservative Q-Learning (CQL) (Kumar et al., 2020), and TD3+BC (Fujimoto & Gu, 2021), focus on cloning expert behaviors and may be adversely affected by the sub-optimal behaviors present in the dataset (Bai et al., 2022).

While more recent action-based support set approaches, such as Bootstrapping Error Accumulation Reduction (BEAR) (Wu et al., 2019a) and Supported Policy Optimization (SPOT) (Wu et al., 2022), attempt to relax cloning conditions through supported regularization, they still face the challenge of being overly restrictive when learning from non-expert offline data. Specifically, they may reject all actions not taken by the behavior policy (i.e., Out of Distribution, OOD actions), including those that, while potentially unsafe, could lead to in-distribution safe states.

In this paper, we proposed a new method to address the above issues. Our key idea is to design a reward mechanism based on whether the subsequent states by following the current policy are beneficial to improve the performance (e.g., being safe - falling within the state support area), instead of explicitly restricting the range of action space for each state. In other words, in our method, the OOD actions are *allowed* as long as they are safe and are beneficial to performance improvement. In this way, our approach is not only more conducive to shaping the desired behavior but also

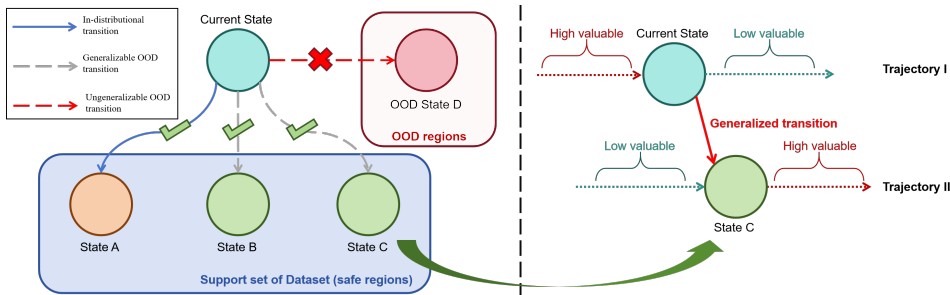

Figure 1: This figure showcases how our ODAF method excels in "trajectory stitching". Specifically, in the left diagram, our method bases decisions on whether a transition's outcomes are backed by the dataset. Even if the necessary action is unsupported by the dataset, its safe outcomes suggest potential for generalization, warranting inclusion in the candidate set. The right diagram illustrates the benefits of this decision-making approach, seamlessly combining high-value segments from diverse trajectories through generalization of unseen transitions, resulting in superior trajectories.

less susceptible to being misled by sub-optimal behavior policies. This is in contrast with the aforementioned action-support constraints-based offline reinforcement learning algorithms (e.g., SPOT, SVR, et al.), which overlooks the correlation between agent decision-making and potential outcomes, thus diminishing the agent's flexibility in decision-making.

In particular, our method focuses on the potential consequences that an action can yield, rather than the specific properties of the action itself, e.g., whether it looks like the behavior policy. Actually, there are previous methods that can be seen through this lens. For example, State Deviation Correction (SDC) (Zhang et al., 2022) and Out-of-sample Situation Recovery (OSR) (Jiang et al., 2023), which were initially developed to help agents recover from Out-of-distribution (OOD) situations by trying to align the transition behavior of the learned policy with that of the behavior policy, can be thought of as matching the consequences of the actions with those in the dataset. However, even these methods are not robust against non-expert data as they do not take the quality of decisions' consequences into account. Actually, blindly cloning the transitions in the dataset may hinder the process of "trajectory stitching" in the case of a sub-optimal behavior policy. As illustrated in Figure 1, the optimal trajectory cannot be synthesized by either OSR or SDC methods, while our proposed method, called **O**utcome-**D**riven **A**ction **F**lexibility (**ODAF**), naturally tolerates unseen transitions during the "trajectory stitching" process, thereby enhancing the agent's ability to learn from non-expert data.

In what follows, after an introduction and a review of related works, Section 3 provides a brief overview of the preliminary knowledge on action constraint methods and consequence-driven methods in offline RL. Section 4 details the ODAF method, along with a theoretical explanation of its effect and the practical implementation. Experimental results are presented in Section 5 to evaluate the effectiveness of both methods under various settings. Finally, the paper concludes with a summary.

## 2 RELATED WORKS

**Action-supported offline RL.**    Action-supported regularization plays a pivotal role in offline RL, striking a balance between conservatism and the ability of the learned policy to stitch trajectories. The Bootstrapping Error Accumulation Reduction (BEAR) (Wu et al., 2019a) method pre-trains an empirical behavior policy and regulates the divergence within a relaxation factor of the new policy. Supported Policy Optimization (SPOT) (Wu et al., 2022) takes a different approach by explicitly estimating the behavior policy's density using a high-capacity Conditional VAE (CVAE) (Kingma & Welling, 2014) architecture. The most recent advancement in this field is Supported Value Regularization (SVR) (Mao et al., 2023), which simplifies action-supported regularization by only requiring an estimation of the behavior policy's action visitation frequency, significantly reducing estimation errors and enhancing robustness. However, action-supported regularization would be too restrictive in avoiding all unseen actions, even those with safe consequences and are worth exploring.

**State recovery-based offline RL.**    As the relaxation of action-constraint, state recovery-based methods like State Deviation Correction (SDC) (Zhang et al., 2022) align the transitioned distributions

of the new policy and the behavior policy, forming a robust transition to avoid the OOD consequences. To further avoid the explicit estimation of consequences in high-dimensional state space, Out-of-sample Situation Recovery (OSR) (Jiang et al., 2023) introduces an inverse dynamics model (IDM) (Allen et al., 2021) to consider the consequential knowledge in an implicit way when decision making. In this paper, we also consider them as the consequence-driven methods that implicitly avoid the *state distributional shift* problem via aligning the transitioned distribution of the new policy with that of the behavior policy. But such methods would hinder their ability of *trajectory stitching* and generalization on non-expert data.

**Trajectory stitching in offline RL.** Recently, Model-based Return-conditioned Supervised Learning (MBRCSL) (Zhou et al., 2023) is proposed to equip the agent with trajectory stitching ability. Although this method has achieved great improvement in certain scenarios, demonstrating the importance of *trajectory stitching*, it needs a large number of rollouts with the pre-trained model to correct the sub-optimal data distribution of the dataset, accumulating the model error. This motivates us to propose the ODAF method to achieve the *trajectory stitching* ability via only policy constraint.

## 3 PRELIMINARIES

A reinforcement learning problem is usually modeled as a Markov Decision Process (MDP), which can be represented by a tuple of the form $(S, A, P, R, \gamma, \rho_0)$, where $S$ is the state space, $A$ is the action space, $P$ is the transition probability matrix, $R$ and $\gamma$ are the reward function and the discount factor, $\rho_0$ is the initial state distribution. A policy is defined as $\pi : S \rightarrow A$ that makes decisions acting with the environment.

In general, we define a Q-value function $Q^\pi(s, a) = (1 - \gamma)\mathbb{E}[\sum_{t=0}^{\infty} \gamma^t R(s_t, \pi(a_t|s_t))|s, a]$ to represent the expected cumulative rewards. Besides, we define the advantage as $A^\pi(s, a) = Q^\pi(s, a) - V^\pi(s)$, where $V^\pi(s) = \mathbb{E}_{a \sim \pi(\cdot|s)}[Q^\pi(s, a)]$. Then we define the $\gamma$-discounted future state distribution (stationary state distribution) for convenience as, $d^\pi(s) = (1 - \gamma)\sum_{t=0}^{\infty} \gamma^t Pr(s_t = s; \pi, \rho_0)$, where $\rho_0$ is the initial state distribution and the $(1 - \gamma)$ is the normalization factor.

In offline setting, Q-Learning (Watkins & Dayan, 1992) learns a Q-value function $\hat{Q}(s, a)$ and a policy $\pi$ from a dataset $\mathcal{D}$ collected by a behavior policy $\pi_\beta$, which consists of quadruples $(s, a, r, s') \sim d^{\pi_\beta}(s)\pi_\beta(a|s)P(r|s, a)P(s'|s, a)$. Then the objective is minimizing the Bellman error over the offline dataset (Watkins & Dayan, 1992), using exact or an approximate maximization scheme, such as CEM (Kalashnikov et al., 2018), onto the above method to recover the greedy policy,

$$\min_Q \mathbb{E}_{(s,a,r,s') \sim \mathcal{D}}\big[r + \gamma\mathbb{E}_{s' \sim P(s'|s,a)}[\mathbb{E}_{a' \sim \pi(a'|s')}Q(s', a')] - Q(s, a)\big]^2 \tag{1}$$

$$\max_\pi \mathbb{E}_{s \sim \mathcal{D}}\mathbb{E}_{a \sim \pi(\cdot|s)}[Q(s, a)] \tag{2}$$

### 3.1 ACTION-SUPPORTED OFFLINE RL

The well-known extrapolation error problem would occur (Fujimoto et al., 2019) when estimating the $\max_{a'} Q(s', a')$ in the Eq.(1)'s TD target. Methods, such as BEAR (Wu et al., 2019a), SPOT (Wu et al., 2022) and SVR (Mao et al., 2023), are proposed to address such issue while preserving the ability of trajectory stitching through the action-supported regularization. In general, these methods could be represented in the following form,

$$\min_Q \mathbb{E}_{(s,a,r,s') \sim \mathcal{D}}\big[r + \gamma\mathbb{E}_{s' \sim P(s'|s,a)}[\max_{\pi \in \Pi_{ac}} \mathbb{E}_{a' \sim \pi(\cdot|s')}Q(s', a')] - Q(s, a)\big]^2 \tag{3}$$

$$\Pi_{ac} = \{\pi | \forall s, supp(\pi(a|s)) \subseteq supp(\pi_\beta(a|s))\} \tag{4}$$

where the $\Pi_{ac}$ is the candidate policy set, in which all the policies $\pi$ would only generate actions supported by the behavior policy $\pi_\beta$.

## 4 THE METHOD

In this section, we introduce the proposed method in detail. First, the objective of the proposed Outcome-Driven Action Flexibility (ODAF) is given and its properties are discussed in a theoretical

way. Then we give the way of implicit implementation, where we utilize the uncertainty lower bound of Q ensembles to approximate the ODAF constraint, which is utilized for empirical analysis in Sec.5.

## 4.1 OUTCOME-DRIVEN POLICY BOOTSTRAPPING

First, we define an outcome-driven candidate set for policy search,

$$\Pi = \{\pi | \forall s \in \mathcal{D}, supp(P(s'|s, \pi)) \subseteq supp(d^{\pi_\beta}(s'))\} \tag{5}$$

where $supp(p)$ denotes the support set of a distribution $p$, $d^{\pi_\beta}$ denotes the stationary state distribution of the behavior policy $\pi_\beta$, and $P(s'|s, \pi) = \mathbb{E}_{a \sim \pi(\cdot|s)} P(s'|s, a)$ is the transitioned distribution of the new policy $\pi$. In words, this $\Pi$ defines a policy set for a given environment based on some behavior policy $\pi_\beta$, in which each policy is safe in the sense that by following it, the transition state will always fall within the support of $d^{\pi_\beta}$. Comparing to previous methods, e.g., those defined in E.q.(4), we see that our candidate policy set $\Pi$ are based on the outcome of the policy rather than the behaviors the policy performed.

In fact, compared to action-support candidate sets as in Eq.(4), this outcome-driven policy candidate set imposes a looser constraint on the policy space, which ensures that under the optimal state coverage assumption (Xie et al., 2021) (this will be discussed later), the optimal policy will fall into our set. Therefore, intuitively, the Bellman operator constructed using this candidate set is expected to have better theoretical properties.

Despite the aforementioned advantages, finding an optimal solution from the policy set defined in Eq. (5) is a computationally challenging problem. In what follows in this section, we will construct a formal theoretical framework to address this issue. Specifically, we first define the Outcome-Driven bootstrapping Bellman operator over $\Pi$ as follows:

$$\hat{T}^\Pi Q(s, a) := r(s, a) + \gamma \mathbb{E}_{s' \sim \hat{P}(s'|s, a)} \max_{\pi \in \Pi} \mathbb{E}_{a \sim \pi(a|s)} Q(s, a) \tag{6}$$

where $\hat{P}$ is the empirical dynamics model of the dataset. In particular, if using the true dynamics model $P$ to replace $\hat{P}$, the $\hat{T}^\Pi$ would be noted as $T^\Pi$.

To justify the above Outcome-Driven bootstrapping Bellman operator, Theorem 1 gives the performance lower bound of the value function learned by this Bellman operator. Before Theorem 1, Lemma 1 shows that the Outcome-Driven bootstrapping Bellman operator is a $\gamma$-contraction operator.

**Lemma 1.** *(Contraction.) The Bellman operator defined in Eq.(6) is a contraction operator.*

The proof of Lemma 1 is shown in Appendix A.

**Theorem 1.** *If we have constructed the outcome-driven policy candidate set $\Pi$, such that the transitioned distribution of all the candidate policies are covered by the dataset well, i.e., $\forall \pi \in \Pi, s \in \mathcal{D}, \sup_{s'} \frac{\hat{P}(s'|s, \pi)}{d^{\pi_\beta}(s')} \leq \epsilon < 1$. Then we can bound the performance lower bound of our method,*

$$\|\hat{Q}^k - Q^*\|(s, a) \leq \frac{\gamma R_{max}}{1 - \gamma} \sqrt{\frac{2}{N \cdot d^{\pi_\beta}(s, a)} \log(\frac{|S||A| \cdot 2^{|S|}}{\delta})} + \gamma^k \cdot \epsilon \cdot \|\triangle_0\| \tag{7}$$

*where $|S|, |A|$ are the dimensions of the state and action spaces. $\|\triangle_0\| = \max_{\pi \in \Pi}(\hat{Q}^0 - \hat{Q}^*)$, where $\hat{Q}_0$ is an arbitrary initial value function and $\hat{Q}^*$ is the fixed point of $\hat{T}^\Pi$, and $Q^*$ is the fixed point of $T^\Pi$. $R_{max}$ is the upper bound of rewards and $N$ is the size of dataset.*

The proof of Theorem 1 is shown in Appendix A.

**Corollary 1.** *If we assume the dataset has sufficient coverage over the optimal policy's stationary state distribution, i.e., $\sup_s \frac{d^{\pi^*}(s)}{d^{\pi_\beta}(s)} \leq C$, then the fixed point $Q^*$ of $T^\Pi$ would be the true optimal value function of the MDP. Then Theorem 1 can bound the learned agent's performance lower bound.*

Theorem 1 and Corollary 1 indicates that the convergence of the value function learned by the Outcome-Driven bootstrapping Bellman operator constructed in Eq.(5) is influenced by the data number $N \cdot d^{\pi_\beta}(s, a)$ and the quality of the approximate solution to the candidate set of state-supported policies (see Eq.(5)). Therefore, we ought to determine how to design a policy constraint

algorithm that maximizes the satisfaction of the outcome-driven constraints proposed in Eq.(5) during the learning process, which would be introduced in detail in the next section.

Before ending this section, we provide two discussions to briefly compare the performance lower bound of our method with other methods, including the action-support constraint method (Wu et al., 2019a) and the pessimistic constraint method (Xie et al., 2021).

**Discussion 1.** *(vs. Action-supported constraint) Assuming we learn the action support set with the same accuracy as in the assumption of Theorem 1, i.e., $\sup_{s,a} \frac{\pi(a|s)}{\pi_\beta(a|s)} \leq \epsilon$. Consequently, we can derive the following, $\forall s \in \mathcal{D}$,*

$$\pi(a|s) \leq \pi_\beta(a|s) \cdot \epsilon \Rightarrow P(s'|s,\pi) \leq P(s'|s,\pi_\beta) \cdot \epsilon \Rightarrow \frac{\mathbb{E}_{d^{\pi_\beta}(s)} P(s'|s,\pi)}{d^{\pi_\beta}(s')} \leq \epsilon \qquad (8)$$

*Then we have $\exists s \in \mathcal{D}$, s.t., $\frac{P(s'|s,\pi)}{d^{\pi_\beta}(s')} > \epsilon$, which conflict the assumption for Theorem 1. Therefore, Theorem 1 could not be applied onto the action-supported method as in (Wu et al., 2019a).*

*As far as we know, to bound the performance lower bound of the action-support methods, it is important to raised the centralization assumption as in the paper (Wu et al., 2019a), which often requires sufficient coverage of the entire state-action space by the dataset. This is challenging to achieve in practice.*

**Discussion 2.** *(vs. Pessimistic constraint) First, we introduce the optimal coverage assumption in (Xie et al., 2021), i.e., $\sup_s \frac{d^{\pi^*}(s)}{d^{\pi_\beta}(s)} \leq C$. Then we have $supp(d^{\pi^*}(s)) \subseteq supp(d^{\pi_\beta}(s))$, so $\pi^* \in \Pi_{ODAF}$, where $\Pi_{ODAF}$ is a state-supported candidate set as defined in Eq.(5).*

*Next, we can directly apply Corollary 2 from (Xie et al., 2021). This corollary is particularly general, as it does not impose any prior assumptions about the policy candidate set, only requiring that the optimal policy is included within that set. Denoting the policy learned by ODAF as $\hat{\pi}_{ODAF}$,*

$$J(\pi^*) - J(\hat{\pi}_{ODAF}) \leq \mathcal{O}\left( \frac{V_{max}\sqrt{C_2}}{1-\gamma} \sqrt{\frac{\log \frac{|F||\Pi_{ODAF}|}{\delta}}{N}} + \frac{\sqrt{C_2}(\epsilon_{F,F} + \epsilon_F)}{1-\gamma} \right) \qquad (9)$$

*where $J(\pi) = \mathbb{E}_{\rho_0(s)} V^\pi(s)$ is the performance of $\pi$ with initial distribution $\rho_0$. $V_{max}$ is the upper bound of value function, $|F|$ and $|\Pi_{ODAF}|$ are the sizes of the approximated Q function space and the policy candidate set, $\epsilon_{F,F}$ and $\epsilon_F$ bound the quality of the learned q function, as are assumed in (Xie et al., 2021).*

*We observe that, since the size of our policy candidate set $|\Pi_{ODAF}|$ is smaller than that of the Pessimistic method in (Xie et al., 2021), which almost has no constraint over the policy candidate set, the bound of our method must be tighter than that of the Pessimistic method.*

*It is worth noting that due to the assumption $\sup_s \frac{d^{\pi^*}(s)}{d^{\pi_\beta}(s)} \leq C$ could not guarantee that the optimal policy is in the action-supported policy candidate set $\Pi_{ac} = \{\pi | \forall s \in \mathcal{D}, supp(\pi(a|s)) \subseteq supp(\pi_\beta(a|s))\}$ as in Eq.(4), the bound in Eq.(9) can not apply to the action-supported methods.*

### 4.2 An Uncertainty-based regularization Algorithm for Implementation

To find the best policy in $\Pi$, we can utilize an $\epsilon-$approximation for the policy support set $\Pi \approx \Pi_\epsilon$, where $\Pi_\epsilon = \{\pi | \forall s \in \mathcal{D}, s', s' \notin supp(d^{\pi_\beta}(s')) \Rightarrow P(s'|s,\pi) < \epsilon\}$. Then with a Lagrange approximation performed on the regularization $\pi \in \Pi_\epsilon$, we have the following objective function[1],

$$\min_\pi \beta_{ssb} \cdot \sum_{s' \notin supp(d^{\pi_\beta}(s'))} (P(s'|s,\pi) - \epsilon) \qquad (10)$$

Then to implement Eq.(10), recall that if we execute an action $a$ at any state $s$ in the offline dataset $\mathcal{D}$, the distribution of the potential consequence $s'$ is given by the dynamics model $P(s'|s,a)$. Our key idea is then to approximate Eq.(10) from above based on the estimation of the state uncertainty of the state $s'$ resulted from a policy. We call it Outcome-Driven Action Flexibility (ODAF).

---

[1]Note that for simplicity, here we ignore the Bellman error term (see E.q.( 2)).

To this end, we may define a new learning objective by minimizing the state uncertainty of the new policy $\pi$ over the perturbed $s$, as follows,

$$\min_{\pi} \mathbb{E}_{s\sim\mathcal{D}} \left[ \sum_{s'} P(s'|s,\pi)U^{\pi}(s') \right] \tag{11}$$

where $U^{\pi}(s) = \mathbb{E}_{\pi(a|s)} U(s,a)$ and $U(s,a) \in [0,+\infty)$ is an uncertainty quantifier as is introduced in (Jin et al., 2021a), which has proven to have the property that if the input data $(s,a)$ is OOD, the $U(s,a)$ would be large and otherwise it would be small (An et al., 2021). Here we utilize it as the indicator to judge the averaged reliability of the learned policy $\pi$ over the potential consequences, aiming to margin out those behaviors which would lead to unsafe consequences.

Next we explicitly build the connection between the uncertainty-based regularization in Eq.(11) and the support region of the dataset. In particular, Theorem 2 shows that under certain mild condition given in Appendix B, we can use uncertainty to bound the support region of the dataset.

**Theorem 2.** *Given an arbitrary state $s$, a conservative policy $\pi$ and a state estimator $U^{\pi}$ based on the policy $\pi$. Then the minimizing the ODAF term in Eq.(11), i.e.,*

$$\min_{\pi} \sum_{s'} P(s'|s,\pi)U^{\pi}(s')$$

*is equivalent to minimizing the upper bound of the following objective as in Eq.(10),*

$$\sum_{s' \notin supp(d^{\pi_\beta}(s'))} P(s'|s,\pi) \tag{12}$$

*where $supp(d^{\pi_\beta}(s'))$ is the support of the dataset.*

Proof of Theorem 2 is given in Appendix C.1. By Theorem 2 , we see that it is less likely for the agent to select the actions that would transit to those states outside the support region of the dataset, hence avoiding being misled by the sub-optimal behavior data, as what may happen for a naive behavior cloning algorithm. [2]

In practice, we implement the proposed Outcome-Driven Action Flexibility (ODAF) onto a SAC-N (An et al., 2021) framework. The ODAF in Eq.(11) could be implemented as the loss,

$$L_{odaf} = \mathbb{E}_{s\sim\mathcal{D}} \left[ \max_{\hat{s}\in\mathbb{B}_s^{\epsilon_{odaf}}} \left[ \sum_{\hat{s}'} P(\hat{s}'|\hat{s},\pi)U^{\pi'}(\hat{s}') \right] \right] \tag{13}$$

where $B_s^{\epsilon_{odaf}}$ is a perturbation ball around state $s$ with magnitude $\epsilon_{odaf}$. The learned policy $\pi'$ is soft-updated via the new policy $\pi$ in this implementation. Here we implicitly assume that the $\mathbb{B}_s^{\epsilon_{odaf}}$ term is related to $\pi$ in that the state $s$ is sampled from the latter's state occupancy $d^{\pi}(\cdot)$, where the implementation of $\mathbb{B}_s^{\epsilon_{odaf}}$ is discussed in Appendix E.1. In words, the new objective Eq.(11) aims to find a robust policy $\pi$ that minimizes the maximum (worst) possibility of driving the agent to encounter unfamiliar regions. *Such regularization could also be theoretically referred as a relaxed state recovery principle (discussed in detail in Appendix D, due to the limitation of space).*

Here we select the standard deviation based uncertainty estimator in (Bai et al., 2022; An et al., 2021):

$$U^{\pi}(s) \approx \beta \cdot Std(Q^k(s,a)) = \beta \cdot \sqrt{\frac{1}{K}\sum_{k=1}^{K}\left(Q^k(s,a) - \bar{Q}(s,a)\right)} \tag{14}$$

where $\{Q^k\}_{k=1}^K$ is the Q-ensemble and $\beta$ is a constant. The Eq.(13) often utilize a Monte-Calro approximation in implementation. Then we attach the $L_{odaf}$ in Eq.(13) onto the actor loss introduced in (Bai et al., 2022) as,

$$L_{\pi} = -\mathbb{E}_{s\sim\mathcal{D},a\sim\pi(\cdot|s)}\left[ \min_{j=1..N} Q'_j(s,a) - \beta\log\pi(a|s) \right] + \beta_{odaf} \cdot L_{odaf} \tag{15}$$

where $\beta_{odaf}$ is the wight of the ODAF term. The critic loss function $L_Q$ is as introduced in (Bai et al., 2022),

$$L_Q = \mathbb{E}_{s,a,r,s'\sim\mathcal{D}}\left[ \left(Q(s,a) - (r+\gamma\mathbb{E}_{a'\sim\pi(\cdot|s')}\left[\min_{j=1..N} Q'_j(s',a') - \beta\log\pi(a'|s')\right])\right) \right] \tag{16}$$

where $\{Q'_j\}_{j=1}^K$ are the $K$ Q ensembles.

To sum up, the whole ODAF could be implementation as Algorithm 1.

---

[2] Empirical evidences that ODAF term is adequate for our needs are also given in Appendix 5.5.

---

**Algorithm 1** The pseudocode of the proposed Uncertainty-based Outcome-Driven Action Flexibility (ODAF) algorithm

---

**Input**: the offline dataset $\mathcal{D}$, maximal update iterations $T$, the pretrained dynamics model $P$.
**Parameter**: policy network $\pi$, evaluation policy $\pi'$, Q-networks $Q_i$.
**Policy Training**
   Initialize the policy network, Q-networks.
   **while** $t < T$ **do**
      Sample mini-batch of N samples $(s, a, r, s')$ from $\mathcal{D}$.
      Get the perturbed $\hat{s}$ with adversarial method in (Yang et al., 2022).
      Feed $\hat{s}$ to the policy network $\pi$ and get $\hat{a}$.
      Feed $\hat{s}, \hat{a}$ to the dynamics model $P$ and get the potential consequence $\hat{s}'$.
      Compute the agent's state uncertainty of $\hat{s}'$ according to the Q-networks $Q$ and $\pi'$.
      Update the policy network $\pi$ according to Eq.(15).
      Update the Q-networks according to (16).
      Soft update the parameters of the evaluation policy $\pi'$.
   **end while**

---

**Output:** The learned policy network $\pi$.

---

## 5 EXPERIMENTS

In experiments we mainly aim to answer the following three key questions:

1) Does ODAF achieve the state-of-the-art performance on standard MuJoCo benchmarks with non-expert data, compared to the latest closely related methods?

2) Does ODAF has better stability (generalization ability) when learning on non-expert data?

3) Does ODAF enable the agent to stitch the sub-optimal trajectories to achieve higher performance?

Our experimental section is organized as follows: First, we perform a test on PointMaze benchmark - a benchmark designed especially for testing the agent's *trajectory stitching* (Zhou et al., 2023) to directly confirm whether our method can achieve our claim - stitching for better trajectories and getting higher performance, answering Question 3; then, we verify that it is hard for the traditional methods to learn from non-expert datasets on standard MuJoCo benchmarks, but the proposed method ODAF has a superior performance among these methods, answering Question 1; to answer Question 2, we perform ODAF in the MuJoCo with limited valuable data setting (Zhang et al., 2022; Jiang et al., 2023; Mao et al., 2023) to explore how the performance of ODAF changes when learning with different levels of non-expert data; besides, we also conduct the experiments over the more complicated tasks of AntMaze to evaluate the ability of multi-step dynamic programming; finally, we conduct a validation study to verify what role the ODAF term plays. [3]

### 5.1 POINTMAZE: TRAJECTORY STITCHING TESTING

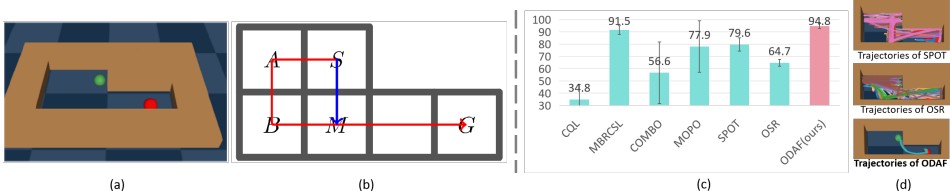

Figure 2: Sub-figure (a) shows the PointMaze map we used. (b) shows the dataset description, where S is the initial point and G is the goal. The red line is a sub-optimal trajectory while the blue line is a trajectory for stitching. (c) shows the results of the methods, and the proposed ODAF is marked red and the highest score is bolded. (d) is the visualization of part of results in (c).

To investigate if the learned agents could do stitching, we introduce a specially designed PointMaze dataset (Zhou et al., 2023), which consists of two kinds of sub-optimal trajectories with equal number,

---

[3]A brief introduction of our code is available in Appendix E.4. *Some experiments are listed in Appendix E.*

as is shown in (a) and (b) of Figure 2: 1) A detour trajectory S → A → B → G that reaches the goal in a sub-optimal manner; 2) A trajectory for stitching: S → M, whose return is very low, but is essential for getting the optimal policy. The optimal trajectory should be a stitching of the two trajectories in dataset (S → M → G). **The resulting dataset has averaged return 40.7 and highest return 71.8**.

To answer question 3, we compare the proposed ODAF with several offline RL baselines, including: 1) Traditional action-constraints: CQL (Kumar et al., 2020) and SPOT (Wu et al., 2022); 2) Consequence-driven method: OSR (Jiang et al., 2023); 3) Model-based methods: COMBO (Yu et al., 2021) and MOPO (Yu et al., 2020); 4) The method specially designed for trajectory stitching: MBRCSL (Zhou et al., 2023). The results are shown in (c) of Figure 2, where the ODAF and MBRCSL both outperforms all the other baselines with a large margin, successfully stitching together sub-optimal trajectories. However, unlike MBRCSL, our method ODAF does not need large number of rollouts based on the approximated dynamics model, which means that ODAF is less likely to suffer from the error accumulation of the learned model, achieving higher performance and better efficiency.

In (d) of Figure 2, we observe that the trajectories generated via SVR and OSR are scattered and coincide with the trajectories listed in the dataset, which demonstrates that these two methods would significantly over-fit to the transitions listed in the dataset instead of generalizing to those unseen but with higher value. However, the proposed ODAF successfully generate trajectories stitched with the two kinds of samples demonstrated in the dataset, achieving higher performance.

### 5.2 LEARNING FROM NON-EXPERT DATASETS

In this section, we compare the two proposed implementations of our method with several significant methods, including CQL (Kumar et al., 2020), PBRL (Bai et al., 2022), SPOT (Wu et al., 2022), SVR (Mao et al., 2023), EDAC (An et al., 2021), RORL (Yang et al., 2022), SDC (Zhang et al., 2022) and OSR-10 (Jiang et al., 2023), based on the D4RL (Fu et al., 2020) dataset in the standard MuJoCo benchmarks. **MuJoCo (D4RL).** There are three types of high-dimensional control environments representing different robots in D4RL: Hopper, Halfcheetah and Walker2d, and five kinds of datasets: 'random', 'medium', 'medium-replay', 'medium-expert' and 'expert'. The 'random' is generated by a random policy and the 'medium' is collected by an early-stopped SAC (Haarnoja et al., 2018) policy. The 'medium-replay' collects the data in the replay buffer of the 'medium' policy. The 'expert' is produced by a completely trained SAC. The 'medium-expert' is a mixture of 'medium' and 'expert'.

Table 1: Results of **ODAF(ours)**, CQL, PBRL, SPOT, SVR, EDAC, RORL, SDC and OSR-10 on offline MuJoCo tasks averaged over 4 seeds. We bold the highest scores in each task.

| | | CQL | PBRL | SPOT | SVR | EDAC | RORL | SDC | OSR-10 | ODAF(Ours) |
|---|---|---|---|---|---|---|---|---|---|---|
| halfcheetah | r | 17.5 | 11.0 | 35.3 | 27.2 | 28.4 | 28.5 | **36.2** | 26.7 | 30.2±1.7 |
| | m | 47.0 | 57.9 | 58.4 | 60.5 | 65.9 | 66.8 | 47.1 | 67.1 | **68.7±0.3** |
| | m-e | 75.6 | 92.3 | 86.9 | 94.2 | 106.3 | 107.8 | 101.3 | 108.7 | **111.1±2.4** |
| | m-r | 45.5 | 45.1 | 52.2 | 52.5 | 61.3 | 61.9 | 47.3 | 64.7 | **65.1±0.3** |
| | e | 96.3 | 92.4 | 97.6 | 96.1 | 106.8 | 105.2 | 106.6 | 106.3 | **107.9±1.1** |
| hopper | r | 7.9 | 26.8 | 33.0 | 31.0 | 25.3 | 31.4 | 10.6 | 30.4 | **32.1±1.5** |
| | m | 53.0 | 75.3 | 86.0 | 103.5 | 101.6 | 104.8 | 91.3 | 105.5 | **106.3±1.2** |
| | m-e | 105.6 | 110.8 | 99.3 | 111.2 | 110.7 | 112.7 | 112.9 | 113.2 | **114.3±0.8** |
| | m-r | 88.7 | 100.6 | 100.2 | 103.7 | 101.0 | 102.8 | 48.2 | 103.1 | **104.8±0.8** |
| | e | 96.5 | 110.5 | 112.3 | 111.1 | 110.1 | 112.8 | 112.6 | 113.6 | **114.7±0.7** |
| walker2d | r | 5.1 | 8.1 | 21.6 | 2.2 | 16.6 | 21.4 | 14.3 | 19.7 | **24.4±2.3** |
| | m | 73.3 | 89.6 | 86.4 | 92.4 | 92.5 | 102.4 | 81.1 | 102.0 | **104.1±2.8** |
| | m-e | 107.9 | 110.8 | 112.0 | 109.3 | 114.7 | 121.2 | 105.3 | 123.4 | **123.8±0.7** |
| | m-r | 81.8 | 77.7 | 91.6 | 95.6 | 87.1 | 90.4 | 30.3 | 93.8 | **95.1±1.9** |
| | e | 108.5 | 108.3 | 109.7 | 110.0 | 115.1 | 115.4 | 108.3 | 115.3 | **115.9±1.3** |
| average | | 67.4 | 74.4 | 78.8 | 80.0 | 82.9 | 85.7 | 70.2 | 86.2 | **87.9** |

The results is shown in Table 1, where part of the results for the comparative methods are obtained by (Yang et al., 2022; Jiang et al., 2023). We have observed that the performance of all methods experiences a significant decrease when applied to non-expert datasets such as 'random', 'medium', 'medium-replay', and 'medium-expert'. This highlights the inherent difficulty in learning from non-expert data in practical settings. However, our proposed method, ODAF, consistently outperforms

other approaches across most benchmarks, particularly surpassing methods that rely on behavior cloning such as CQL, PBRL, and EDAC. Furthermore, ODAF achieves state-of-the-art performance in terms of the average score. Additionally, we would like to emphasize that ODAF demonstrates significant improvements over the state-of-the-art conservative methods (e.g., SVR and OSR) on the 'medium' and 'medium-replay' datasets. This notable margin can be attributed to ODAF's ability to avoid error compounding through its flexibility in trajectory stitching. This further underscores the advantages of ODAF in effectively handling non-expert data. In the next section, we will delve deeper into exploring the advantages of ODAF across different levels of non-expert datasets.

## 5.3 INFLUENCE OF DIFFERENT LEVELS OF NON-EXPERT DATA

In this section, we further explore the feasibility of the proposed ODAF on different levels of non-expert offline dataset, where we mix the 'expert' and 'random' datasets with different ratios. This is a setting widely used, such as in (Zhang et al., 2022; Mao et al., 2023; Jiang et al., 2023). Here, the proportions of 'random' data are 0.5, 0.6, 0.7, 0.8 and 0.9, for Halfcheetah, Hopper and Walker2d.

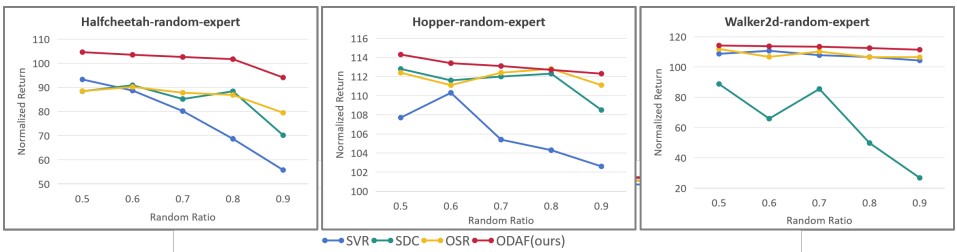

Figure 3: The results on the MuJoCo benchmarks with different levels of non-expert data.

We compare the proposed ODAF with SVR (Mao et al., 2023), OSR (Jiang et al., 2023) and SDC (Zhang et al., 2022), as is shown in Figure 3, ODAF outperforms the other three methods on three type of control environments over the normalized scores. We have observed that both of our proposed methods, particularly ODAF, exhibit a significantly lower decrease rate over the 'Halfcheetah' benchmark compared to the other two methods as the random ratio increases, which can be attributed to the agent's heightened sensitivity to the quality of data collection in this environment. Furthermore, when testing on the 'Hopper' and 'Walker2d' benchmarks, we note that ODAF demonstrates the least decrease in performance among all methods when the random ratio reaches 0.9, which highlights the advantage of the implicit implementation in addressing more complex tasks and learning from data of lower quality in practical scenarios. Therefore, we emphasize that our method, ODAF, is better equipped for learning with non-expert data, and they exhibit improved stability and performance across various benchmarks with lower data quality.

## 5.4 EXPERIMENTS ON MORE COMPLICATED ENVIRONMENT - ANTMAZE

Table 2: Results of **ODAF(ours)**, CQL, IQL, SPOT, ATAC, SDC and OSR-10 on offline AntMaze tasks averaged over 4 seeds. We bold the highest scores in each task.

| | | CQL | IQL | SPOT | ATAC | SDC | OSR-10 | ODAF(Ours) |
|---|---|---|---|---|---|---|---|---|
| AntMaze | umaze | 82.6 | 87.5 | 93.5 | 70.6 | 89.0 | 89.9 | **94.6±0.9** |
| | umaze-diverse | 10.2 | 62.2 | 40.7 | 54.3 | 57.3 | **74.0** | 71.3±4.7 |
| | medium-play | 59.0 | 71.2 | 74.7 | 72.3 | 71.9 | 66.0 | **79.0±2.1** |
| | medium-diverse | 46.6 | 70.0 | 79.1 | 68.7 | 78.7 | **80.0** | 79.6±1.7 |
| | large-play | 16.4 | 39.6 | 35.3 | 38.5 | 37.2 | 37.9 | **59.3±5.7** |
| | large-diverse | 3.2 | **47.5** | 36.3 | 43.1 | 33.2 | 37.9 | 47.4±9.3 |
| average | | 36.3 | 63.0 | 59.9 | 57.9 | 61.2 | 64.3 | **71.9** |

Compared to the MuJoCo environment, the AntMaze environment requires the agent to have the ability of multi-step dynamic planning, making it considered a more complex scenario. In this environment, we compare CQL (Kumar et al., 2020), IQL (Kostrikov et al., 2022), SPOT (Wu et al., 2022), ATAC (Cheng et al., 2022), SDC (Zhang et al., 2022), and OSR-10 (Jiang et al., 2023). In

the AntMaze environment, based on the size and shape of the maze, it can be categorized into 'umaze,' 'medium,' and 'large'; and based on different tasks, it can be classified as 'diverse' and 'play'. From the results in Table 2, we can observe that our method outperforms other methods in most environments, particularly in the 'large' and 'diverse' tasks, where our method significantly outperforms others. This indicates that our method exhibits strong generalization capabilities even when facing more complex and challenging tasks.

### 5.5 VALIDATION STUDY FOR ODAF REGULARIZATION

In this section, we perform a series of validation experiments to explore the impact of two key components of the proposed method: the pre-trained dynamics models $\hat{P}(s'|s, a)$ and the uncertainty approximations $U^\pi(s')$. Both components are integrated into the ODAF term in Eq. (11), which evaluates the safety of the outcome resulting from a given action. To assess the effectiveness of the ODAF term, we conducted a straightforward experiment within the MuJoCo environment.

In the experiment, we first generated two sets of actions: one set with safe outcomes, obtained by selecting two similar states from the dataset and generating actions through the inverse dynamics model; the other set with unsafe outcomes, composed of a series of random actions. We then utilized either the true dynamics model (TDM) or our pre-trained dynamics model (PDM) to predict the next states of these actions and assess their safety as $score(s, a) = \mathbb{E}_{\hat{P}(s'|s,a)} U^\pi(s')$.

Table 3: Validation study for ODAF regularization.

|  | Halfcheetah | Hopper | Walker2d |
|---|---|---|---|
| Safe actions w. TDM | 0.07 | 0.13 | 0.12 |
| Random actions w. TDM | 0.48 | 0.49 | 0.53 |
| Safe actions w. PDM | 0.13 | 0.11 | 0.21 |
| Random actions w. PDM | 0.54 | 0.64 | 0.57 |

Table 3 shows the results. Comparing the results of the first and second rows, we observe that our safety scoring method is sensitive to whether the consequences of actions are in-distribution or out-of-distribution (OOD), which supports the validity of this measurement. Looking at the results of the third and fourth rows, we find that the uncertainty quantifier also reveals a significant score gap between the two types of actions when using the pre-trained dynamics model. This gap is comparable to that observed in the first and second rows. This suggests that the performance of the pre-trained dynamics model is sufficient to distinguish whether the consequences of actions are safe, without even requiring a perfect reconstruction of the outcome state of those actions.

## 6 CONCLUSION

In this paper, we propose a novel method called Outcome-Driven Action Flexibility (ODAF) to trade-off the conservatism and generalization when learning from non-expert data in offline RL. In particular, ODAF liberates the agent from the shackles of non-expert data in a consequence-driven manner - it implicitly avoids the agent suffering from *distributional shift* via controlling its consequences within in-distributional (safe) regions, while preserving its ability of trajectory stitching, which is critical for achieving superior performance from non-expert demonstrations. We provide two ways to implement the proposed idea. Theoretical and experimental results validate the effectiveness and feasibility of ODAF. We believe that the problem addressed in this work and the proposed method hold promise for practical applications of offline RL.

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

# Appendix

## A PROOF OF LEMMA 1 AND THEOREM 1.

**Lemma 1. (Contraction.)** *The Bellman operator defined in Eq.(6) is a contraction operator.*

*Proof.* Suppose there exist two variables $u, v$ in the value function space, then we have,

$$\|T^\Pi u - T^\Pi v\|_\infty = \max_s |T^\Pi u(s) - T^\Pi v(s)| \tag{17}$$

$$= \max_s |\max_{\pi \in \Pi} \mathbb{E}[r + \gamma u(s')|s, a] - \max_{\pi \in \Pi} \mathbb{E}[r + \gamma v(s')|s, a]| \tag{18}$$

$$\leq \max_s \max_{\pi \in \Pi} |\gamma \mathbb{E}[u(s') - v(s')|s, a]| \tag{19}$$

$$\leq \max_s \max_{\pi \in \Pi} \gamma \mathbb{E}[|u(s') - v(s')| \big| s, a] \tag{20}$$

$$\leq \max_s \max_{\pi \in \Pi} \gamma \mathbb{E}[\|u(s') - v(s')\|_\infty \big| s, a] \tag{21}$$

$$= \gamma \|u(s') - v(s')\|_\infty \tag{22}$$

Completing the proof.

**Theorem 1.** *If we have constructed the policy candidate set $\Pi$, such that the transitioned distribution of all the candidate policies are covered by the dataset well, i.e., $\forall \pi \in \Pi, s \in \mathcal{D}$, $\sup_{s'} \frac{\hat{P}(s'|s,\pi)}{d^{\pi_\beta}(s')} \leq \epsilon < 1$. Then we can bound the performance lower bound of our method,*

$$\|\hat{Q}^k - Q^*\|(s, a) \leq \frac{\gamma R_{max}}{1 - \gamma} \sqrt{\frac{2}{N \cdot d^{\pi_\beta}(s, a)} \log(\frac{|S||A| \cdot 2^{|S|}}{\delta})} + \gamma^k \cdot \epsilon \cdot \|\triangle_0\| \tag{23}$$

*where $|S|, |A|$ are the dimensions of the state and action spaces. $\|\triangle_0\| = \max_{\pi \in \Pi}(\hat{Q}^0 - \hat{Q}^*)$, where $\hat{Q}_0$ is an arbitrary initial value function and $\hat{Q}^*$ is the fixed point of $\hat{T}^\Pi$, and $Q^*$ is the fixed point of $T^\Pi$. $R_{max}$ is the upper bound of rewards and $N$ is the size of dataset.*

*Proof of Theorem 1.* First we decompose the $\|\hat{Q}^k - Q^*\|(s, a) = \|\hat{Q}^k - \hat{Q}^*\|(s, a) + \|\hat{Q}^* - Q^*\|(s, a)$ with the triangle inequality.

Then we aim to bound $\|\hat{Q}^* - Q^*\|$. First, by the triangle inequality, we have,

$$\|\hat{Q}^* - Q^*\|(s, a) \leq \|\hat{T}^\Pi \hat{Q}^* - \hat{T}^\Pi Q^*\|(s, a) + \|\hat{T}^\Pi Q^* - Q^*\|(s, a) \tag{24}$$

Because $\hat{T}^\Pi$ is a $\gamma$-contraction operator (see Lemma 1), we have,

$$\|\hat{Q}^* - Q^*\|(s, a) \leq \frac{\|T^\Pi Q^* - T^\Pi Q^*\|(s, a)}{1 - \gamma} \tag{25}$$

$$= \frac{\gamma}{1 - \gamma} \|P(s'|s, a) - \hat{P}(s'|s, a)\|_1 \max_{\pi \in \Pi} Q^*(s', \pi) \tag{26}$$

$$\leq \frac{\gamma \cdot R_{max}}{1 - \gamma} \|P(s'|s, a) - \hat{P}(s'|s, a)\|_1 \tag{27}$$

$$\leq \frac{\gamma \cdot R_{max}}{1 - \gamma} \sqrt{\frac{2}{N \cdot d^{\pi_\beta}(s, a)} \log(\frac{|S||A| \cdot 2^{|S|}}{\delta})} \tag{28}$$

The last inequality holds because of the **Proposition 9** in Ghavamzadeh et al. (2016).

Then we bound the $\|\hat{Q}^k - \hat{Q}^*\|(s, a)$.

$$\|\hat{Q}^k - \hat{Q}^*\|(s, a) \leq^{(a)} \gamma^k \|\mathbb{E}_{\hat{P}(s_k|\{\pi_1...\pi_{k-1}\}, s, a)} \max_{\pi \in \Pi}(\hat{Q}^0 - \hat{Q}^*)(s_k, \pi)\| \tag{29}$$

$$= \gamma^k \|\sum_{s_k} \frac{\hat{P}(s_k|\{\pi_1...\pi_{k-1}\}, s, a)}{d^{\pi_\beta}(s_k)} d^{\pi_\beta}(s_k) \max_{\pi \in \Pi}(\hat{Q}^0 - \hat{Q}^*)(s_k, \pi)\| \tag{30}$$

$$\leq \gamma^k \sum_{s_k} \frac{\hat{P}(s_k|\{\pi_1...\pi_{k-1}\}, s, a)}{d^{\pi_\beta}(s_k)} \|d^{\pi_\beta}(s_k) \max_{\pi \in \Pi}(\hat{Q}^0 - \hat{Q}^*)(s_k, \pi)\| \tag{31}$$

$$\leq \gamma^k \sup_t \sum_{s_t} \frac{\hat{P}(s_t|\{\pi_1...\pi_{t-1}\}, s, a)}{d^{\pi_\beta}(s_t)} \sum_{s_k} \|d^{\pi_\beta}(s_k) \max_{\pi \in \Pi}(\hat{Q}^0 - \hat{Q}^*)(s_k, \pi)\| \tag{32}$$

$$= \gamma^k \sup_t \sum_{s_t} \frac{\hat{P}(s_t|\{\pi_1...\pi_{t-1}\}, s, a)}{d^{\pi_\beta}(s_t)} \|\triangle_0\| \tag{33}$$

$$\tag{34}$$

where $\forall i \in [1, k], \pi_i = \arg\max_{\pi \in \Pi}(\hat{Q}^{k-i} - \hat{Q}^*)(s, \pi)$. The k-step transition distribution $\hat{P}(s_k|\{\pi_1...\pi_k\}, s, a)$ means that starting form $s, a$, taking policy from index $1...k$, and the final state distribution at the k-step.

The inequality $(a)$ holds because,

$$\|\hat{Q}^k - \hat{Q}^*\|(s, a) = \|\hat{T}^\Pi \hat{Q}^{k-1} - \hat{T}^\Pi \hat{Q}^*\|(s, a) \tag{35}$$

$$= \gamma \|\mathbb{E}_{\hat{P}(s_1|s, a)}(\hat{Q}^{k-1} - \hat{Q}^*)(s_1, \pi_1)\| \tag{36}$$

$$\leq \gamma \|\mathbb{E}_{\hat{P}(s_1|s, a)}(\gamma \mathbb{E}_{\hat{P}(s_2|\pi_1, s_1)}(\hat{Q}^{k-2} - \hat{Q}^*)(s_2, \pi_2))\| \tag{37}$$

$$= \gamma^2 \|\mathbb{E}_{\hat{P}(s_2|\pi_1, s, a)}(\hat{Q}^{k-2} - \hat{Q}^*)(s_2, \pi_2)\| \tag{38}$$

$$........... \tag{39}$$

$$= \gamma^k \|\mathbb{E}_{\hat{P}(s_k|\{\pi_1...\pi_{k-1}\}, s, a)}(\hat{Q}^0 - \hat{Q}^*)(s_k, \pi_k)\| \tag{40}$$

Then $\forall \pi \in \Pi, s \in \mathcal{D}, \sup_{s'} \frac{\hat{P}(s'|s, \pi)}{d^{\pi_\beta}(s')} \leq \epsilon$, so we have, $\forall t, \sup_{s_t} \frac{\hat{P}(s_t|s, \{\pi_1...\pi_t\})}{d^{\pi_\beta}(s_t)} \leq \epsilon^t \leq \epsilon$, where $\pi_1...\pi_t \in \Pi$. Therefore, finally we have,

$$\|\hat{Q}^k - \hat{Q}^*\|(s, a) \leq \gamma^k \cdot \epsilon \cdot \|\triangle_0\| \tag{41}$$

Completing the proof.

# B ASSUMPTION FOR THEOREM 2

**Assumption 1.** *(**Bounded uncertainty**).* $\forall(s, a) \notin supp(d^{\pi_\beta}(s'))$, $U_{min} \leq u(s, a)$, *where* $U_{min}$ *and* $U_{max}$ *are constants, and* $U_{min} > 0$.

where $\mathcal{D}$ is the offline dataset and $supp(D)$ is the support of $\mathcal{D}$. Assumption 1 assumes that for any OOD state-action pair, the uncertainty estimator is strictly positive, which conform to the empirical results in (An et al., 2021).

# C PROOFS

## C.1 PROOF OF THEOREM 2

The proof of Theorem 2 is performed under Assumption 1.

**Theorem 2.***Given an arbitrary state $s$, a conservative policy $\pi$ and a state estimator $U_{\mathcal{D}}^\pi$ based on the policy $\pi$ and dataset $\mathcal{D}$. Then the minimizing the ODAF term in Eq.(11), i.e.,*

$$\min_\pi \sum_{s'} P(s'|s, \pi) U_{\mathcal{D}}^\pi(s')$$

*is equivalent to minimizing the upper bound of the following objective,*

$$\sum_{s' \notin supp(d^{\pi_\beta}(s'))} P(s'|s,\pi) \tag{42}$$

*where $supp(d^{\pi_\beta}(s'))$ is the support of the dataset.*

*Proof.*

$$\sum_{s'} P(s'|s,\pi)U_{\mathcal{D}}^{\pi}(s')$$

$$= \sum_{s' \in supp(d^{\pi_\beta}(s'))} P(s'|s,\pi)U_{\mathcal{D}}^{\pi}(s') + \sum_{s' \notin supp(d^{\pi_\beta}(s'))} P(s'|s,\pi)U_{\mathcal{D}}^{\pi}(s') \tag{43}$$

$$\geq \sum_{s' \notin supp(d^{\pi_\beta}(s'))} P(s'|s,\pi)U_{\mathcal{D}}^{\pi}(s') \tag{44}$$

where, by Assumption 1, then $\forall s' \in \mathcal{D}, U_{\mathcal{D}}^{\pi}(s') > 0$, so that Eq.(43) upper bounds Eq.(44). Then, via Assumption 1, we have,

$$\sum_{s' \notin supp(d^{\pi_\beta}(s'))} P(s'|s,\pi)U_{\mathcal{D}}^{\pi}(s')$$

$$= \sum_{s' \notin supp(d^{\pi_\beta}(s'))} P(s'|s,\pi)\mathbb{E}_{a'\sim\pi(\cdot|s')}u(s',a')$$

$$\geq \sum_{s' \notin supp(d^{\pi_\beta}(s'))} P(s'|s,\pi)\mathbb{E}_{a'\sim\pi(\cdot|s')}U_{min}$$

$$= U_{min} \cdot \sum_{s' \notin supp(d^{\pi_\beta}(s'))} P(s'|s,\pi)$$

where $U_{min}$ is a constant according to $\pi$. Therefore we have,

$$\min_{\pi} U_{min} \cdot \sum_{s' \notin supp(d^{\pi_\beta}(s'))} P(s'|s,\pi)$$

$$\Leftrightarrow \min_{\pi} \sum_{s' \notin supp(d^{\pi_\beta}(s'))} P(s'|s,\pi)$$

Complete the proof. □

## D A LINK TO TRADITIONAL CONSEQUENCE-DRIVEN METHODS

In this section, we give an interesting link between the proposed ODAF and the SDC (Zhang et al., 2022) (or OSR (Jiang et al., 2023)). First, we would like to introduce a weight KL divergence as is in (Moharana & Kayal, 2019) as $D_{KL}^{w}(p(x)\|q(x)) = \sum_x w(x)p(x)\log\frac{p(x)}{q(x)}$, where $w(x)$ is the weight factor. First, a smooth assumption should be given,

**Assumption 2.** *(Smooth). Given the new policy $\pi$ and the behavior policy $\pi_\beta$. For any state $s \sim \mathcal{D}$ and its perturbed state $\hat{s} \sim \mathbb{B}_s^{\epsilon_{odaf}}$, the ratio of the transitioned distributions of these two policies is bounded, i.e., $\delta = \max_{s'} \frac{P(s'|s,\pi_\beta)}{P(s'|\hat{s},\pi)}$.*

Then in particular,

**Theorem 3.** *Given a state $s \sim \mathcal{D}$ and the perturbed state $\hat{s} \in \mathbb{B}_s^{\epsilon_{odaf}}$, the log-prob. version of ODAF term in Eq.(11) upper bounds the equivalent form of a weighted version of model-based OSR term as in Eq.(??) with a term that maximizes the entropy of the transitioned distribution,*

$$\min_{\pi} \max_{\hat{s} \in \mathbb{B}_s^{\epsilon_{odaf}}} D_{KL}^{w}(P(s'|s,\pi_\beta)\|P(s'|\hat{s},\pi)) - \delta \cdot H(P(\cdot|\hat{s},\pi)) \tag{45}$$

*where $D_K^w L(p\|q)$ is a weighted KL divergence, and the weight factor $w(s') = \frac{U_{max} - U_{\mathcal{D}}^\pi(s')}{U_{max}} \in [0, 1]$. The $\delta$ is the upper bound of the transitioned distributions' ratio as in Assumption 2 and the $U_{max}$ is the upper bound of the uncertainty estimator.*

*Proof.* First we attach a log probability trick onto the ODAF term in Eq.(11). Given a state $s \sim \mathcal{D}$ and the perturbed state $\hat{s} \in \mathbb{B}_s^{\epsilon_{odaf}}$, we have

$$\min_\pi \max_{\hat{s} \in \mathbb{B}_s^{\epsilon_{odaf}}} \sum_{s'} U_{\mathcal{D}}^\pi(s') \cdot \log P(s'|\hat{s}, \pi) \tag{46}$$

Then we have,

$$\min_\pi \max_{\hat{s} \in \mathbb{B}_s^{\epsilon_{odaf}}} \sum_{s'} \log P(s'|\hat{s}, \pi) U_{\mathcal{D}}^\pi(s')$$

$$\Leftrightarrow \min_\pi \max_{\hat{s} \in \mathbb{B}_s^{\epsilon_{odaf}}} \sum_{s'} \log P(s'|\hat{s}, \pi) U_{\mathcal{D}}^\pi(s')$$

$$+ \delta \cdot U_{max} \cdot H(P(\cdot|\hat{s}, \pi)) - \delta \cdot U_{max} \cdot H(P(\cdot|\hat{s}, \pi))$$

$$\geq^{(a)} \min_\pi \max_{\hat{s} \in \mathbb{B}_s^{\epsilon_{odaf}}} \sum_{s'} \log P(s'|\hat{s}, \pi) U_{\mathcal{D}}^\pi(s') + \sum_{s'} U_{max} P(s'|s, \pi_\beta) \log \frac{1}{P(\hat{s}'|\hat{s}, \pi)} - \delta \cdot U_{max} \cdot H(P(\cdot|\hat{s}, \pi))$$

$$\geq^{(b)} \min_\pi \max_{\hat{s} \in \mathbb{B}_s^{\epsilon_{odaf}}} \sum_{s'} [U_{max} - U_{\mathcal{D}}^\pi(s')] P(s'|s, \pi_\beta) \log \frac{1}{P(s'|\hat{s}, \pi)} - \delta \cdot U_{max} \cdot H(P(\cdot|\hat{s}, \pi))$$

$$\geq^{(c)} \min_\pi \max_{\hat{s} \in \mathbb{B}_s^{\epsilon_{odaf}}} [U_{max} - U_{\mathcal{D}}^\pi(s')] \cdot \sum_{s'} P(s'|s, \pi_\beta) \log \frac{P(s'|s, \pi_\beta)}{P(s'|\hat{s}, \pi)} - \delta \cdot U_{max} \cdot H(P(\cdot|\hat{s}, \pi))$$

$$\Leftrightarrow \min_\pi \max_{\hat{s} \in \mathbb{B}_s^{\epsilon_{odaf}}} \sum_{s'} w(s') \cdot P(s'|s, \pi_\beta) \log \frac{P(s'|s, \pi_\beta)}{P(s'|\hat{s}, \pi)} - \delta \cdot H(P(\cdot|\hat{s}, \pi))$$

$$\Leftrightarrow^{(d)} \min_\pi \max_{\hat{s} \in \mathbb{B}_s^{\epsilon_{odaf}}} D_{KL}^w(P(s'|s, \pi_\beta)\|P(s'|\hat{s}, \pi)) - \delta \cdot H(P(\cdot|\hat{s}, \pi)) \tag{47}$$

Note that the inequality $(a)$ holds because $\delta = \max_{s'} \frac{P(s'|s, \pi_\beta)}{P(s'|\hat{s}, \pi)}$.

And the inequality $(b)$ depends on $\forall s, s'$, the probability $P(s'|s, \pi_\beta) \leq 1$.

Then the inequality $(c)$ holds via adding a negative constant $[U_{max} - U_{\mathcal{D}}^\pi(s')]P(s'|s, \pi_\beta) \log P(s'|s, \pi_\beta) < 0$ according to the policy $\pi$.

Finally the equivalence $(d)$ holds via the definition of weighted KL divergence $D_{KL}^w(p(x)\|q(x)) = \sum_x w(x)p(x) \log \frac{p(x)}{q(x)}$, as in (Moharana & Kayal, 2019).

Complete the proof. □

Theorem 3 demonstrates that minimizing the log-prob. version of ODAF term in Eq.(11) is equivalent to minimizing the upper bound of a weighted OSR while maximizing the entropy of the transitioned distribution. The weight $w(s')$ takes into account the qualitative characteristic related to state uncertainty, and the maximization of the entropy of the transitioned distribution $P(\cdot|\hat{s}, \pi)$ ensures sufficient variance of the policy a prior. Theorem 3 shows that, compared to OSR-10, our method further takes the quality of the consequences of decisions into account, which means that even if the consequences of actions fall within the state support set, the agent may not be rewarded if there is high uncertainty. In this sense, our method is more robust to challenging tasks and sub-optimal offline data compared to the OSR-10 approach.

# E  EXTERNAL EXPERIMENTS

## E.1  ADVERSARIAL ATTACKS

We adopt three attack methods, namely random, action diff, and min Q following prior works (Zhang et al., 2020; Pinto et al., 2017), about which the details are discussed in (Yang et al., 2022), as follows,

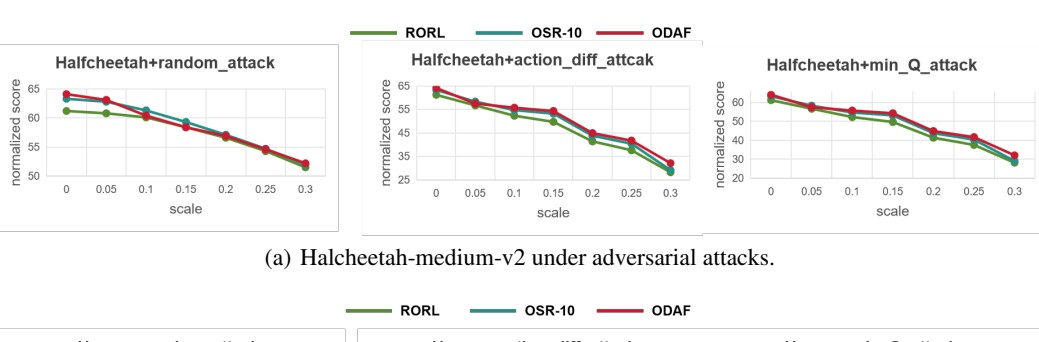

(a) Halcheetah-medium-v2 under adversarial attacks.

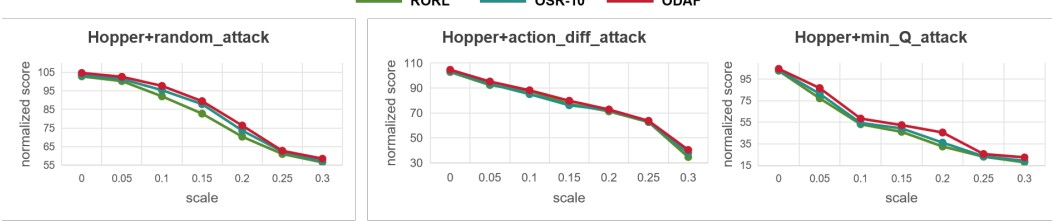

(b) Hopper-medium-v2 under adversarial attacks.

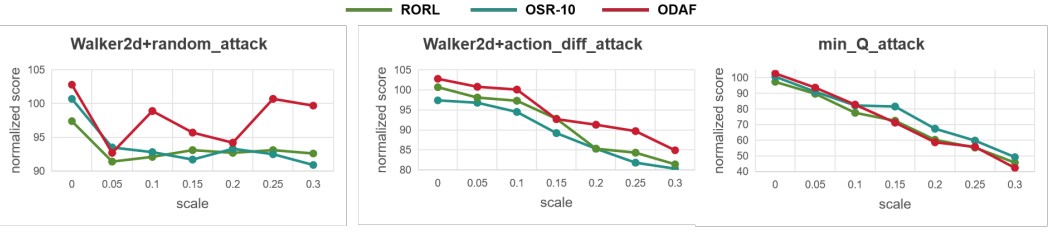

(c) Walker2d-medium-v2 under adversarial attacks.

Figure 4: Results of RORL, OSR-10, ODAF (ours) on MuJoCo under different scales of adversarial attacks on the normalized score.

1) **random**: uniformly samples perturbed states in an $l_\infty$ ball of norm $\epsilon$.

2) **action_diff**: an effective attack based on the agent's policy and is proved to be an upper bound on the performance difference between perturbed and unperturbed environments (Zhang et al., 2020). It directly finds perturbed states in an $l_\infty$ ball of norm $\epsilon$ to satisfy $\max_{\hat{s} \in \mathbb{B}_d(s,\epsilon)} D_J(\pi(\cdot|s)\|\pi(\cdot|\hat{s}))$.

3) **min_Q**: an attack requires both the agent's policy and value function to perform a relatively stronger attack. The attacker finds a perturbed state to minimize the expected return of taking an action from that state: $\min_{\hat{s} \in \mathbb{B}_d(s,\epsilon)} Q(s, \pi(\hat{s}))$. For ensemble-based algorithms, Q is set as the mean of ensemble Q functions.

The results of RORL, OSR-10 and our method ODAF are show in Figure 4, from which we observe that the proposed ODAF outperforms other two methods on most of the benchmarks, especially at the benchmarks with 'action_diff' attacks, which obtain the perturbed state with the largest divergence with the learned policy, probably because ODAF is more skilled in dealing with the OOD states where the agent's decision making is sensitive to the perturbations. However, we also remark that ODAF is not good at dealing with 'min_Q' attacks, probably due to ODAF's decision making depends on the ability of the Q-ensemble, which would be a valuable direction for research.

### E.2 SENSITIVE ANALYSIS OVER HYPERPARAMETERS OF ODAF

The ODAF weight $\beta_{odaf}$ is the hyperparameter that control the magnitude of how the ODAF term influence the training. Its influence to ODAF is as shown in Figure 5, where three agents are all trained on the 'meidum' datasets. From the results we remark that the best choice for $\beta_{odaf}$ in this implementation is around 0.3 in the three standard MuJoCo benchmarks.

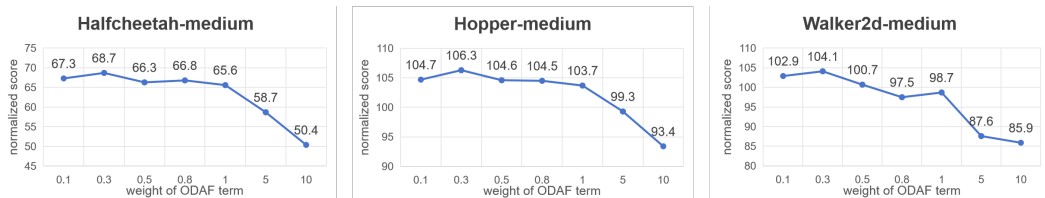

Figure 5: The sensitive analysis results of $\beta_{odaf}$. The left is on 'halfcheetah-medium', the middle is on 'hopper-medium' and the right is on 'walker2d-medium'.

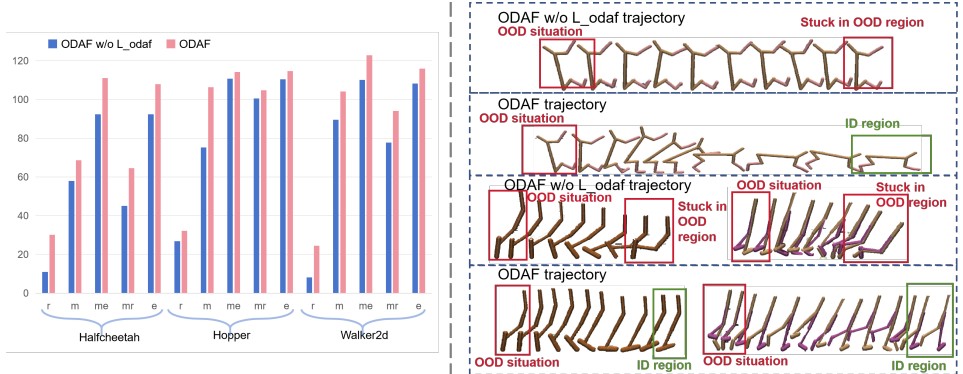

Figure 6: The results of ODAF and ODAF w/o $L_{odaf}$ on the standard MuJoCo benchmarks.

### E.3 ABLATION STUDY

In this section, we perform an ablation study on the two implementations to evaluate how the ODAF term behaves. From the results in the left part of Figure 6, we observe that ODAF significantly outperforms ODAF w/o $L_{odaf}$ (the regularization term in Eq.(15)) by nearly 20% improvement on average, which demonstrates the important role ODAF term playS in learning a higher-capacity policy that is more likely to control the agent moving within the high-valuable regions.

Furthermore, we also visualize some of the results of ODAF and ODAF w/o $L_{ssb_u}$ on the 'Halfcheetah-OOS-large', 'Hopper-OOS-large' and 'Walker2d-OOS-large' benchmarks, as shown in the right part of Figure 6, from which we observe that compared with the results of ODAF w/o $L_{ssb_u}$, the ODAF agent generalizes better when falling into OOD situations and is more likely to generate transitions with those in-distributional consequences, enhancing the robustness, which could also be seen as a phenomenon that follows the (loosed) *state recovery* principle in another way.

### E.4 CODE

We build the proposed based on the RORL project from github[4]. The reasons why we choose YangRui2015's project are as follows: 1) The RORL framework is a classic baseline for the conservative offline reinforcement learning based on an implementation of PBRL (Bai et al., 2022). 2) Learning conservative Q functions can be easily implemented using the RORL framework. 3) To our knowledge, the RORL framework is the baseline with the highest scores in MuJoCo benchmarks. Our code is provided in the supplemental material.

### E.5 TRAINING DETAILS

In this section, we introduce our training details, including: 1) the hyperparameters our method use; 2) the structure of the neural networks we use: the Q-networks, inverse dynamics model network and policy network; 3) the training details of ODAF; 4) the total amount of compute and the type of resources used.

---

[4]Project of RORL: https://github.com/YangRui2015/RORL

### E.5.1 HYPERPARAMETERS OF ODAF

In Table 4 and Table 5, we give the hyperparameters used by ODAF to generate Table 1 results. The $\epsilon_{odaf}$ is the perturbation scalar of a perturbation ball $B_s^{\epsilon_{odaf}}$ around state $s$ in ODAF loss and $\beta_{odaf}$ is the weight of the ODAF loss.

Table 4: Hyperparameters of ODAF in standard MuJoCo benchmarks.

|                    | **Halfcheetah** | **Hopper** | **Walker2d** |
|--------------------|-----------------|------------|--------------|
| $\epsilon_{odaf}$  | 0.001           | 0.005      | 0.01         |
| $\beta_{odaf}$     | 0.3             | 0.3        | 0.3          |

Table 5: Hyperparameters of ODAF in adversarial attack MuJoCo benchmarks.

|                    | **Halfcheetah** | **Hopper** | **Walker2d** |
|--------------------|-----------------|------------|--------------|
| $\epsilon_{odaf}$  | 0.05            | 0.005      | 0.07         |
| $\beta_{odaf}$     | 0.3             | 0.3        | 0.3          |

### E.5.2 NEURAL NETWORK STRUCTURES OF ODAF

In this section, we introduce the structure of the networks we use in this paper: policy network, Q network and the dynamics model network.

The structure of the policy network and Q networks is as shown in Table 6, where 's_dim' is the dimension of states and 'a_dim' is the dimension of actions. 'h_dim' is the dimension of the hidden layers, which is usually 256 in our experiments. The policy network is a Guassian policy and the Q networks includes ten Q function networks and ten target Q function networks.

Table 6: The structure of the policy net and the Q networks.

| policy net              | Q net                   |
|-------------------------|-------------------------|
| Linear(s_dim, 256)      | Linear(s_dim, h_dim)    |
| Relu()                  | Relu()                  |
| Linear(h_dim, h_dim)    | Linear(h_dim, h_dim)    |
| Relu()                  | Relu()                  |
| Linear(h_dim, a_dim)    | Linear(h_dim, 1)        |

The structure of the dynamics network is as shown in Table 7, which is a conditional variational auto-encoder. 's_dim' is the dimension of states, 'a_dim' is the dimension of actions and 'h_dim' is the dimension of the hidden variables. 'z_dim' is the dimension of the Gaussian hidden variables in conditional variational auto-encoder.

Table 7: The structure of the dynamics model network.

| dynamics model net                                     |
|--------------------------------------------------------|
| Linear(s_dim + a_dim, h_dim)                           |
| Linear(h_dim, h_dim)                                   |
| Linear(h_dim, h_dim)                                   |
| Linear(h_dim, z_dim) │ Linear(h_dim, z_dim)            |
| Linear(s_dim + a_dim + z_dim, h_dim)                   |
| Linear(h_dim, h_dim)                                   |
| Linear(h_dim, s_dim)                                   |

### E.5.3 COMPUTE RESOURCES

We conducted all our experiments using a server equipped with one Intel Xeon Gold 5218 CPU, with 32 cores and 64 threads, and 256GB of DDR4 memory. We used a NVIDIA RTX3090 GPU with 24GB of memory for our deep learning experiments. All computations were performed using Python 3.8 and the PyTorch deep learning framework.

## F DISCUSSION

### F.1 LIMITATIONS

Just like the methods based on the traditional *state recovery* principle, the proposed ODAF is also unable to generalize to those states that are quite far away from the offline dataset, where any action executed would not lead to any low-uncertainty state. In this situation, the ODAF term would not embed any useful information for the new policy, because all the forward consequences have high uncertainty, which make such guidance degrade to a random-walk. Exploring the performance boundary of ODAF is also a major direction for our future work.

### F.2 DIFFERENCES WITH OTHER WORKS

In Figure 7, we illustrate the differences between the basic ideas of our method and other methods, including action-constrained methods (conservative method), action-supported methods and state recovery based methods.

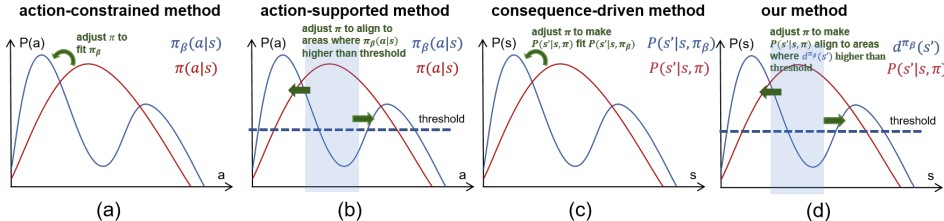

Figure 7: These figures illustrate the fundamental concepts underlying the four types of current conservative constraints. (a) depicts the action-constrained method, which aims to reduce the discrepancy between the new policy and the behavior policy, i.e., behavior cloning. (b) represents the action-supported method, where the new policy generates actions within regions where the density of the behavior policy $\pi_\beta(a|s)$ exceeds a certain threshold. (c) showcases the consequence-driven method, which mimics the behavior policy's transition distribution in the new policy. Lastly, (d) introduces our approach, where the high-density region (blue box) of the new policy's transition distribution $P(s'|s, \pi)$ aligns with areas where the stationary state distribution of the behavior policy $d^{\pi_\beta}(s')$ surpasses the threshold. It is evident that our constraint method is the most relaxed among the four methods, indicating that it may have the potential to achieve better generalization performance.

