# OpenReview forum: "Outcome-Driven Action Flexibility for Robust Offline Reinforcement Learning"
_ICLR.cc/2025/Conference — ICLR 2025 Conference Withdrawn Submission_

### Official Review · Reviewer_oe2t · 2024-10-27

**Soundness:** 2
**Presentation:** 1
**Contribution:** 2
**Rating:** 3
**Confidence:** 3

**Summary:**

The authors introduce ODAF, an offline RL algorithm designed to balance conservatism and flexibility when using realistic non-expert data for training. Unlike action-supported or state recovery-based methods, which either reject all OOD actions or align transition distributions, ODAF introduces an auxiliary reward based on the uncertainty of potential outcomes (i.e., subsequent states).

The authors first propose the Outcome-Driven bootstrapping Bellman operator and verify its convergence. Subsequently, they perform a Lagrange approximation to derive the objective function and its surrogate upper bound. The practical algorithm is implemented by incorporating an outcome uncertainty reward into the actor's objective, building upon the framework of RORL [1].

Comprehensive experiments are conducted to demonstrate the method's effectiveness.

[1] Yang, R., Bai, C., Ma, X., Wang, Z., Zhang, C., & Han, L. (2022). RORL: Robust offline reinforcement learning via conservative smoothing. *Advances in neural information processing systems*, *35*, 23851-23866.

**Strengths:**

1. The "outcome-driven" concept is intriguing, and the comparisons between ODAF, action-supported methods, and pessimistic constraints are insightful.
2. The experimental design, including the detailed appendices, is well-executed, and the empirical results demonstrate the effectiveness of ODAF, achieving 87.9 on all 15 D4RL tasks and approximately 81.7 on 12 non-expert tasks.

**Weaknesses:**

1. The paper's presentation is poor and needs improvement.

2. There are several confusing notational issues, including:

   - In Section 3, Equations 1 and 2 resemble standard online actor-critic optimization objectives, except for the offline dataset notation. Additionally, the text overlaps with Section 3 of [1] and Section 2 of [2].

   - L136 uses $\hat Q$, but $Q$ is used in Equations 1 and 2 instead.

   - L138: As $P$ is defined as the transition probability, it is inappropriate to use $P(r|s,a)$.

   - L186, Equation 6: The last term should be $Q(s', a')$, otherwise it is not a Bellman operator.

   - In the proof of Lemma 1, $u$ and $v$ should be variables in the state-action value space since the operator acts on the $Q$ function.

   - The notations for $U^\pi(s)$ and the perturbation ball $B_s^{\epsilon_{odaf}}$ are inconsistent between Section 4.2 and the Appendix.

    This kind of issues should be double-checked to ensure the correctness and cosistency.

3. The derivation of the optimization objective (Equation 10) is abrupt, and the introduction of the constant $\beta_{ssb}$ lacks explanation. More intermediate steps in deriving Equation 10 should be included in the Appendix

4. Several references are incorrectly formatted, such as:

   - The citation for "RORL: Robust offline reinforcement learning via conservative smoothing" [3], published in NeurIPS 2022, is incomplete.

   - Duplicate references to "*Behavior regularized offline reinforcement learning*" and "*Is pessimism provably efficient for offline RL*" are present.

   - The format is inconsistent for the same conference.

    The reference formatting throughout the paper should be checked carefully, ensuring completeness and consistency

5. Despite the lengthy theoretical proofs, the practical algorithm is relatively simple. As shown in the provided code (in the supplementary material), an additional VAE environment model is pre-trained, and the auxiliary ODAF loss is calculated, with much of the framework relying on RORL [3].

[1] Jiang, K., Yao, J. Y., & Tan, X. (2023). Recovering from out-of-sample states via inverse dynamics in offline reinforcement learning. *Advances in Neural Information Processing Systems*, *36*.

[2] Mao, Y., Zhang, H., Chen, C., Xu, Y., & Ji, X. (2023). Supported value regularization for offline reinforcement learning. *Advances in Neural Information Processing Systems*, *36*.

[3] Yang, R., Bai, C., Ma, X., Wang, Z., Zhang, C., & Han, L. (2022). RORL: Robust offline reinforcement learning via conservative smoothing. *Advances in neural information processing systems*, *35*, 23851-23866.

**Questions:**

1. Why was RORL not compared in the PointMaze and AntMaze tasks?

2. Given the overlap with RORL in practical implementation, the success of ODAF appears to stem largely from the auxiliary loss calculated based on the model. Would other model-based offline RL methods, such as MOREC [1], contribute to further improvements in ODAF?

3. Please see the Weaknesses section for additional detailed questions.

[1] Luo, F. M., Xu, T., Cao, X., & Yu, Y. (2023). In *The Twelfth International Conference on Learning Representations (ICLR 2024)*

**Details Of Ethics Concerns:**

No ethics concerns.

---

### Official Review · Reviewer_V5gn · 2024-10-30

**Soundness:** 4
**Presentation:** 3
**Contribution:** 3
**Rating:** 6
**Confidence:** 4

**Summary:**

The paper proposes a novel method ODAF for offline RL, where the subsequent state distribution is considered to stitch better trajectory. I agree with the authors that constraining the state support looses the constraint on the policy space, thus may find superior policy in the offline dataset. And, the authors conduct sufficient experiments to show the performance of ODAF and verify its effectiveness.

**Strengths:**

* Integrating performance measurement into state uncertainty is a good way to optimise policy.
* The proposed method's theoretical analysis and performance bounds are well-grounded and offer valuable insights into its properties.
* The experiments can effectively illustrate the design of ODAF.

**Weaknesses:**

Since implementing ODAF requires a pretrained dynamics model, I wonder if ODAF has generalization capability with small-size samplings. And if even the pre-trained dynamics model is inaccurate, how about the performance degradation of ODAF?

**Questions:**

The limitation should be discussed in the main paper.

---

### Official Review · Reviewer_aeZA · 2024-10-30

**Soundness:** 3
**Presentation:** 2
**Contribution:** 2
**Rating:** 3
**Confidence:** 5

**Summary:**

This paper presents Outcome-Driven Action Flexibility (ODAF), a new offline RL approach designed to improve robustness and generalization, particularly when learning from sub-optimal data. ODAF shifts the emphasis from constraining actions to evaluating the safety and potential outcomes of those actions, allowing the agent to consider a broader set of actions that still lead to favorable results. The authors suggest that this outcome-focused method better addresses the distributional shift challenges common in offline RL, especially with the less-than-ideal policies often found in real-world datasets. The paper provides theoretical support for ODAF, showing that, under certain conditions, the value function it learns converges to the true optimal value. Empirical results on D4RL and PointMaze demonstrate ODAF’s superior performance over current offline RL methods, especially when working with non-expert data. Notably, ODAF shows the ability to stitch segments from various sub-optimal trajectories into higher-value trajectories, leading to more effective learning outcomes.

**Strengths:**

The authors back ODAF with a theoretical foundation. They establish, through Lemma 1, that the outcome-driven Bellman operator has a contraction property, and they show in Theorem 1 that ODAF provides a lower bound for the learned value function. These results add credibility to the approach

At its core, the paper tackles a real challenge in offline RL: learning from non-expert data. This is particularly relevant since real-world data often lacks the precision of expert demonstrations. ODAF’s performance on benchmarks with varying data quality highlights its ability to handle non-expert data. The authors also underscore the role of trajectory stitching in effective offline RL, making a compelling case that ODAF’s focus on outcomes enables it to combine action segments more effectively than methods that are overly conservative in their action selection.

**Weaknesses:**

Limited discussion of limitations: The paper acknowledges the limitations of ODAF in generalizing to states far from the dataset. However, this discussion is brief and lacks a deeper analysis of the conditions under which ODAF might fail or underperform. Expanding this section with concrete examples and potential mitigation strategies would strengthen the paper.

Implementation details could be more comprehensive: The paper provides a high-level overview of the ODAF algorithm and its implementation using uncertainty quantification techniques. However, more detailed information about the specific choices made in the implementation, such as the uncertainty estimation method, the adversarial attack mechanisms, and the (full) choices of hyperparameters, would be beneficial. This would enhance the reproducibility of the work and allow for a more thorough understanding of the factors influencing ODAF's performance.

Grammatical errors and narration: At times, there are major issues with the grammar and writing of the paper, for instance:
- At line 140, the sentence never ends.
- At line 403, "MuJoCo (D4RL)." is redundant.

Moreover, I found the narration overly complicated, making the paper hard to follow and understand.

Comparison with SOTA trajectory stitching methods: The paper highlights ODAF's ability to perform trajectory stitching and compares it with MBRCSL in the Related Work Section. However, in the Experiments Section, other methods are also considered like SPOT, OSR, and COMBO. I suggest discussing all the baselines in the Related Work Section, each in the related subsection.

Missing experiment information: Demonstrating the results should be consistent within the paper, ie if confidence intervals are provided for one method, they should also be provided for other methods. However, in Tables 1 and 2, confidence intervals are missing for baselines. The same for all methods in Figure 3. Also, why baselines are not consistent throughout the paper, eg in Tables 1 and 2?

**Questions:**

The sensitivity analysis for the ODAF weight is presented in the Appendix. It would be helpful to include this analysis in the main paper and discuss the impact of other hyperparameters on ODAF's performance. Are there specific hyperparameter settings that are particularly crucial for achieving good results, or is ODAF generally robust to variations in hyperparameter values?

The experimental evaluation focuses on continuous control tasks in MuJoCo and PointMaze environments. It would be interesting to see how ODAF performs in other domains of D4RL, such as Maze2D or Adroit. Would the outcome-driven approach need modifications to be effective in those settings?

The paper utilizes uncertainty quantification techniques to implement ODAF. How sensitive is ODAF's performance to the choice of uncertainty estimation method? Have the authors experimented with different uncertainty estimation techniques, and if so, how did those choices impact the results?

The paper acknowledges ODAF's limitation in generalizing to states significantly outside the dataset. Are there potential modifications to the ODAF algorithm or its implementation that could mitigate this limitation and improve its performance in such scenarios?

---

### Official Review · Reviewer_4oyJ · 2024-11-02

**Soundness:** 2
**Presentation:** 1
**Contribution:** 3
**Rating:** 3
**Confidence:** 3

**Summary:**

This paper focuses on offline reinforcement learning (ORL), where a policy $\pi$ is learned from an offline collected set of trajectories using a behavior policy $\pi_{\\beta}$. ORL is sensitive to the dataset coverage of the MDP, i.e., how much of the full set of trajectories in the MDP can be found in the offline dataset. To fix this, prior work broadly suggests restricting the learned policy $\pi$ to be a sub-set of the behavior policy $\pi \\subset \pi_{\\beta}$, and never take actions the behavior policy $\pi_{\\beta}$ does not. This work suggests that such an approach is sub-optimal, and instead constrain the learned policy to never take actions that lead to unseen states. The paper includes a theoretical analysis of the work, along with evaluations on several environments and behavior policies.

**Strengths:**

* The overall idea sounds promising. It can be seen as a relaxation of action-based constraints.
* The paper includes a sizable amount of theoretical analyses.
* The work is compared to many baselines in the evaluation.
* I especially like the experiment in §5.1, as it is easy to reason about.

**Weaknesses:**

While I like the paper overall, there are several issues that made it difficult to recommend for acceptance. I do not aim to dishearten the authors, as the work can be significantly improved as long as these concerns are met.
* **Dense and Unclear Theoretical Analyses**: The theoretical analyses are very dense. I found it difficult to understand what the final goal is in each step. I followed the analyses in §4.1 and by the end I was unsure if corollary 1 is even useful. I did not precisely follow the theoretical analyses after discussion 1 as I was already confused by the prior analyses. Perhaps a clarification by the authors could help here.

  Theorem 1 bounds the distance of the learned Q value (with the contraction in Eq. 6) against the optimal Q value. For this distance to converge to zero (needed for corollary 1), we need $N \rightarrow \infty$ and $d^{\pi_{\beta}}(s, a) > 0$. However, if we have these two conditions, normal Watkins style Q-learning will also converge to the optimal point. In problem settings where ORL is not trivial, there is no full coverage, and the analyses should be on such settings. Please note that the stated condition in corollary 1 $sup_{s} \frac{d^{\pi^*}(s)}{d^{\pi_{\beta}}(s)} \le C$ is not (trivially) enough for reducing the distance in theorem 1 to zero, and in fact, one can construct very simple counter examples where the contraction in Eq. 6 will not lead to the optimal Q values.

   Consider an MDP with three states $s_0, s_1$ and  $s_2$. $s_0$ is the initial state, $s_2$ is the terminal state, and each transition advances the state by 1 $s_i \rightarrow s_{i+1}$ regardless of the action. There are two possible actions at each state, where action 0 induces a rewards of 10 and action 1 induces a reward of -10. Now suppose the behavior policy always choose action 1 for $\pi_{\beta}(\cdot|s_0)$, and action 0 or 1 at random for $\pi_{\beta}(\cdot|s_1)$. The stated operator in Eq. 6 will not converge to the optimal policy.

   I would be very happy to be corrected on these statements, in case I misunderstood.

   My suggestion is to begin this section by providing a high-level outline of what the analyses is supposed to show at each step. It would probably be useful to also separate the analyses from discussions related to the final algorithm in §4.2.
* **Vague evaluation settings**: It is unclear how ODAF was tuned for the experiments in §5. While the hyperparameters are noted in E.5, it seems ODAF was *tuned separately for each environment*, which is odd. It would be best to include how the hyperparameters were tuned, and clarify how they were tuned to each environment separately without test-set tuning.

    Something else that I did not quite understand is why MBRCSL is not compared to for experiments in §5.2 and §5.3. Is there some technical limitation that did not allow it? Considering it is the closest baseline in §5.1, its lack of presence needs to be justified.

* **Inconsistent Interpretation of Results**: The bold font should only be used when some method is outperforming all others by *statistically significant margins*. Table 1 does not state what the range is in the ODAF column (standard deviation, 95th percentile confidence interval, ...), nor does it include it for baselines. Assuming it is 95th percentile confidence, and the confidence interval for baselines is zero (which is surely not true), many results are not statistically significant, e.g., half-cheetah m-e, hopper m, walker 2d m/m-e/m-r/r. In fact, *SVR gets a better score in walker2d m-r*. The same can be said for the medium-diverse or large-diverse rows in Table 2, where baselines outperform ODAF, but ODAF is still highlighted in bold.

Some minor nitpicks:
* Typo in Eq. 6: The expectation term on the RHS does not include $s'$.
* $\delta$ is not stated in Theorem 1: It is not clear what $\delta$ is. It seems the theorem is partly based on prior work, but the authors still need to state what $\delta$ is.
* Typo in Eq. 25: The first term in the RHS should use $\hat{T}$, not $T$.

**Questions:**

The most important questions are these:

* How will the analyses in §4.1 change if the dataset does not have full coverage (which is the ORL problem setting)?
* How was ODAF tuned per environment?
* If you include p95 confidence margins for baselines, on how many results is ODAF statistically significant?

---

### Official Review · Reviewer_FfEj · 2024-11-03

**Soundness:** 2
**Presentation:** 2
**Contribution:** 3
**Rating:** 5
**Confidence:** 4

**Summary:**

This paper proposes Outcome-Driven Action Flexibility (ODAF), which prioritizes the beneficial consequences of actions rather than their similarity to behavior policies. This allows the agent to explore Out-of-Distribution (OOD) actions as long as they lead to safe and effective states, enhancing learning flexibility and robustness against sub-optimal behaviors. The paper details the theoretical framework and practical implementation of ODAF, supported by experimental results demonstrating its effectiveness.

**Strengths:**

- The paper is supported by both theoretical and experimental results.
- The motivation and starting point of this paper are relatively new to my knowledge.
- The empirical performance of the algorithm is good.

**Weaknesses:**

- The motivation of this work is not very clear to me. To address the issue of distributional shift in offline RL, the authors propose substituting previous action-constraint methods with the consequence-driven approach ODAF; however, it employs SAC-N as the base algorithm in its experiments, which has already addressed distribution shift. Therefore, it can be difficult to demonstrate whether the proposed consequence-driven approach ODAF actually resolve the distribution shift.
- The paper claims that the proposed consequence-driven approach, ODAF, has better theoretical properties than action-constraint methods. However, I do not find direct theoretical results supporting this point (for example, better performance guarantee). If I have missed any relevant content, please let me know.
- The assumption 1 for Theorem 2 is not correctly presented. Additionally, there are some typos and formatting errors throughout the paper, which need to be carefully reviewed.
- A recent work [1] also proposes to constrain the consequences (next states) of the trained policy rather than its actions to address distribution shifts, while also taking the quality of decisions’ consequences into account. The idea appears very similar (doesn't matter now) but the algorithms are different. The authors are encouraged to include some comparisons in the final version.
- In Discussion 1, how does Eq.8 lead to the argument in line 228 (Then we have …)? Additionally, doesn’t the fact that the assumption of the action support set does not satisfy that of the consequence set indicate that the consequence set can possibly be stricter?

[1] Offline Reinforcement Learning with OOD State Correction and OOD Action Suppression.

**Questions:**

Please see weakness. I am willing to increase the score if my concerns are addressed.

---

### Note · Authors · 2024-11-19

**Comment:**

We appreciate the efforts of the reviewers and the Area Chair regarding our work. However, after discussions with all co-authors, we regret to inform you that we have decided to withdraw our paper without further rebuttals. We will continue to refine our work based on all of the suggestions provided by you.

Thank you once again for your time and efforts !

**Withdrawal Confirmation:**

I have read and agree with the venue's withdrawal policy on behalf of myself and my co-authors.